# Adaptive Notch Filter in a Two-Link Flexible Manipulator for the Compensation of Vibration and Gravity-Induced Distortion



**Minoru Sasaki** [1,*] **, Joseph Muguro** [2] **, Waweru Njeri** [2] **and Arockia Selvakumar Arockia Doss** [3]

1   Intelligent Production Technology Research & Development Center for Aerospace (IPTeCA), Tokai National Higher Education and Research System, Gifu 501-1193, Japan
2   Center for Robotics and Biomedical Engineering, School of Engineering, Dedan Kimathi University of Technology, Nyeri 657-10100, Kenya
3   Vellore Institute of Technology, School of Mechanical Engineering, Chennai 632014, Tamil Nadu, India
*   Correspondence: sasaki@gifu-u.ac.jp; Tel.: +81-90-6462-0957

**Abstract:** This paper presents a 2-link, 2-DOF flexible manipulator control using an inverse feedforward controller and an adaptive notch filter with a direct strain feedback controller. In the flexible manipulator, transient and residue vibrations inhibit the full potential of the manipulator. Vibrations caused by abrupt changes in the direction of the links are referred to as transient vibrations, whereas residual vibrations occur when the arm takes too long to settle after engaging in the intended task. The feedforward adaptive notch filter will reduce transient vibration caused by the manipulator arm beginning and halting suddenly, while the strain feedback will assure the quick decay of leftover vibrations. Maple, Maplesim, and MATLAB tools were used to model the manipulator and create the inverse controller and adaptive notch filter. The experiments took place in the dSPACE control desk environment. The experimental results of the spectral power of strain resulting from the two strategies are compared. From the results, the adaptive notch filter control had over an 80% improvement in the reduction in resonant frequencies that contribute to vibration. The results confirmed the feasibility of the approach, characterized by very minimal transient vibrations and a quick settling of the end effector.

**Keywords:** inverse controller; two-link flexible manipulator; transient/residue vibration; adaptive filter; strain feedback; three degrees of freedom

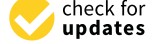



## 1. Introduction

Robots have gained popularity in the 21st century as a means of supplementing and improving quality of life. To date, robots are applied in at least all aspects of human life ranging from space and marine exploration, industrial process automation, transportation, health and wellness, and agriculture, amongst others. Based on the locomotion structure, robots can be broadly categorized as mobile robots (wheeled/legged robots, humanoids, etc.), armed/manipulator robots (robot arm, gantry, etc.), flying robots (remote/autonomous aerial vehicles), underwater robots, or a hybridization generated by combining these areas [1–5]. Each of the categories has constraints and design considerations that are unique to the application. In this paper, we focus on robot arms and review the challenges associated with them.

In a robot arm, one of the design considerations is the tip position tracking. Increasing the structural rigidity will yield a better tracking of the tip position. On the other hand, this will inevitably increase the weight and required materials which leads to increased energy usage during operations and costs. Furthermore, the rigidity and bulkiness of the robot arm bottlenecks the operational speeds of the robot [1,6]. In summary, energy consumption, material and design costs, operation speed, and control schemes, amongst others, are some of the researched themes in robots.

In an attempt to address the challenges, a manipulator is so designed with low-weight, minimal cost materials; this results in the so-called flexible manipulator as contrasted to a rigid-arm robot [7,8]. From the literature, a robot arm constructed using a single link is referred to as a flexible arm while the combination of two or more links is a flexible manipulator [9]. These manipulators have low operational costs due to their use of small actuators. Furthermore, high speeds of operation and a small form factor make the performance of complex tasks possible, thereby fostering their integration in diverse fields.

Flexible manipulators overcome the previously listed challenges, but they introduce new challenges during control, i.e., vibration and deformation. Loading exacerbates the condition, as do every extra joint and linkage. Multiple researchers have focused on vibration and deformation suppression amongst other issues emanating from flexible manipulator arms [10–12]. From a literature review, vibrations were prevalent when the arms were suddenly halted [13]. The intensity of the excited vibrations depends on the velocity prior to this stoppage. From this, input preshaping proposals have been made to arrive at acceleration and deceleration inputs that will excite minimal vibration. Other strategies that are applied in this research touch on infinite dimensional controllers. This includes methods such as strain feedback, robust controllers, and proportional derivative controllers, amongst others.

The authors in [14,15] proposed a direct strain feedback (DSFB) strategy for vibration control. In this approach, a percentage of the root strain is sent back through a fixed gain controller $k$. They demonstrated in their research that DFSB reduces vibrations by boosting system damping without altering the system stiffness. They also experimentally demonstrated that the scheme is asymptotically stable. The essential benefit of DSFB control is its simplicity, as it does not require a plant model for controller design, just joint angles and root strain. On the other hand, this approach has a drawback in that when using fixed gain controllers, noise is feedback when the strain reduces. Furthermore, this control system has a difficulty coping with unexpected parameters and uncertainties such as changes in loads and trajectories.

One of the remedies to the drawback in DFSB is the use of gain tuning as opposed to fixed gain. Several pieces of research have been conducted utilizing various methodologies for tuning the gain [16,17]. In this approach, the tuning process identifies the right proportion to feedback using neural networks or other optimization models. The tuning process relies on the analytical modeling of the manipulator requiring complex computations.

Another approach for vibration control is the use of adaptive notch filter feedforward. This is a control strategy that does not directly need an analytical model but has the effect of altering the spectrum of the target trajectory to generate minimal vibrations. The key challenge of this approach is the adaptation coefficient's slow convergence speed and the filter's stability throughout the adaptation coefficient's convergence process. A two-degrees-of-freedom (DOF) controller with fixed notch filter feedforward control and DSFB control was introduced by authors in [18]. When applied to a 2-link, 3-DOF flexible manipulator, the findings demonstrated the scheme's efficacy. However, because the notch filter's frequency spectrum was constant, when the resonance frequency varied owing to changes in the tip load, the control performance degraded.

As a remedy, in the present paper, we propose a strain feedback mechanism based on passivity-based control to complement this solution. The adaptive notch filter thus used ensures performance with fluctuating tip loads. We constructed the filter based on the simplified lattice algorithm (SLA) to speed up the convergence of the coefficients.

Another significant problem for a multilink flexible manipulator that is initially vertical and then moves in three dimensions is steady-state distortion. In this case, gravity acting on the arm's weight will cause strain values to be inaccurate. The use of such measurements for strain feedback will result in joint angle inaccuracies. There has been little research working towards this area [19–21]. We also propose employing an adaptive notch filter to remove stationary distortion owing to the self-weight of the links.

In the paper, to verify the performance in a variable tip load, we propose using an adaptive notch filter for vibration suppression. To speed up and stabilize the filter in the convergence process, we design an adaptive notch filter based on the simplified lattice algorithm (SLA) as well as the removal of stationary distortion due to the self-weight of the links. We compare the vibration suppression performance of feedforward (FF) control using an adaptive notch filter and DSFB control in a 2-link, 2-DOF flexible arm to verify the effectiveness of suppression. The proposed scheme performed better than the conventional (DSFB) scheme. The improvement was attributed to the adaptation and the self-weight error correction algorithm. The proposed controller was shown to be effective for vibration control and trajectory tracking in situations with variable loads.

The remainder of the paper is laid out as follows. Section 2 introduces the 2-link, 2-DOF flexible manipulator, experiment setup, and adaptive filter design. Section 3 presents and discusses the findings, followed by a conclusion in Section 4.

## 2. Methods

### 2.1. Physical Model Experiment

Figure 1a shows the physical manipulator in use with two links and three degrees of freedom that may be manipulated. The entire connection of boards and signaling is illustrated in Figure 1b. A dSPACE 1103 controller (signal processing) board was used to perform analog-to-digital conversion (ADC) and digital-to-analog conversion (DAC) to support device driving and communication. The motors were driven by a DC motor servo driver which was connected to a dSPACE signal processing board. The motors were equipped with an optical encoder with an output of 1000 pulses/rev. The optical encoders fed back the angles to the servo drive amplifier and into the dSPACE board. Harmonic drives were used to lower the speed of the joints by a factor of 100. Joint 1 rotated perpendicular to the ground (twisting), while joints 2 and 3 rotated in a bending direction. The overall specifications of the model are shown in Table 1.

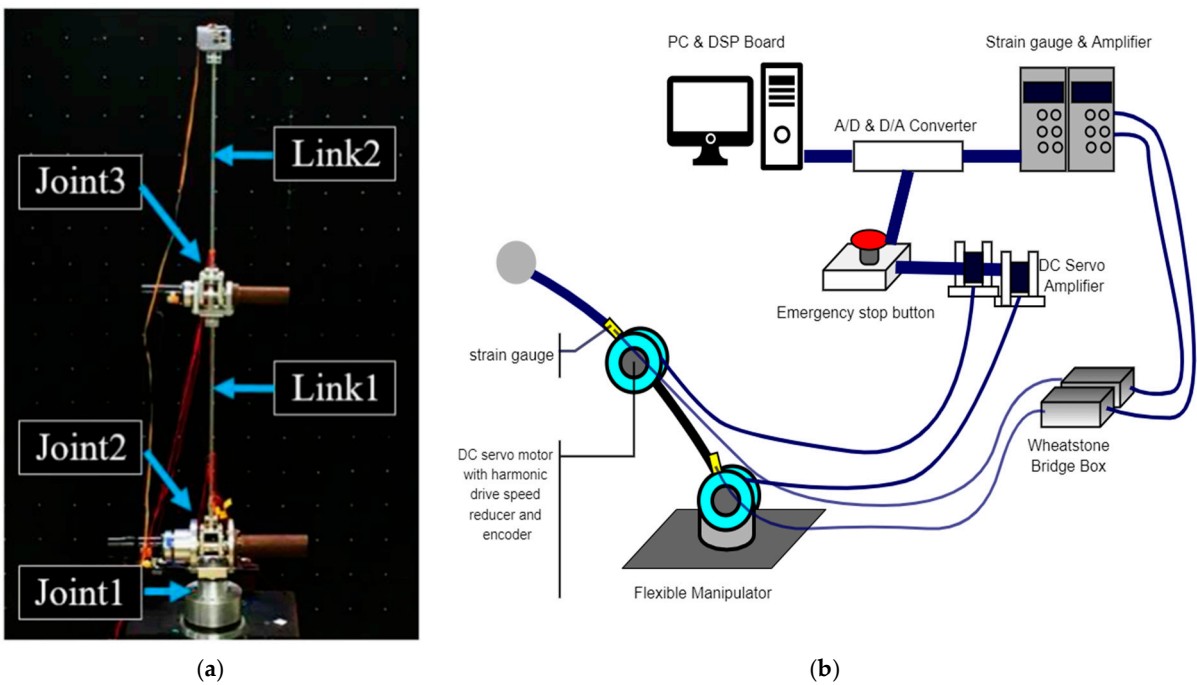

(**a**) (**b**)

**Figure 1.** Setup of the flexible manipulator. (**a**) Actual flexible manipulator. (**b**) Illustration of the manipulator setup.

**Table 1.** Parameter specifications for the flexible manipulator.

| Component | Specifications | | |
|---|---|---|---|
| Motor 1 | Type | V511-012EL8 | units |
| | Rated power | 105 | W |
| | Rated spindle speed | 3000 | Rpm |
| | Rated torque | 0.326 | N·m |
| | Maximum torque | 0.784 | N·m |
| | Mass | 1.1 | Kg |
| Motor 2 | Type | V404-012EL8 | |
| | Rated power | 39 | W |
| | Rated spindle speed | 3000 | Rpm |
| | Rated torque | 0.11 | N·m |
| | Maximum torque | 0.323 | N·m |
| | Mass | 0.55 | kg |
| Encoder | Resolution | 1000 | P/R |
| | Type | Incremental | |
| Reduction Gear Joint 1 | Reduction ratio | 1/100 | |
| | Mass | 0.15 | kg |
| | Type | CSF-40-100-2AR-SP | |
| Reduction Gear Joint 2 | Reduction ratio | 1/100 | |
| | Mass | 0.09 | kg |
| | Type | CSF-17-1002ª-R-SP | |
| Link 1 | Length | 0.44 | m |
| | Radius | 0.005 | m |
| | Material | Stainless | |
| Link 2 | Length | 0.44 | m |
| | Radius | 0.004 | m |
| | Material | Aluminum | |
| Strain Gauge | Type | KGF-2-120-C1-23L1M2R | |

Cast iron cylinders were mounted on the opposite ends of the motor connected to joint 2 and 3. This was carried out to counterbalance motion effects. A weight of 100 g was attached at the tip of the second link of the manipulator which made the link easily deformed during operation. In this case, the joints were considered to be rigid while the links were flexible. Strain gauges were installed at the root of each connection for feedback. The measurement of strain was performed by converting the output strain into electrical resistance and conditioning the output using a bridge circuit before amplification. The amplifiers in use were DPM713B and DPM913B strain amplifiers from Kyowa Electric Co. Ltd. The amplified signal was sampled at 0.02 s and digitized with an ADC by the dSPACE™ DSP board for further processing as shown in Figure 1b.

In the operations of the manipulator for validation and system identification, an actual robot operation was performed, and the data were logged for further analysis. In this case, the manipulator started with a vertical posture at t = 0. A 20 s rectangular pulse input was applied to move the manipulator's tip. During the first 10 s, joints 1, 2, and 3 were fed with the desired angle of 20 degrees which moved link 1 and link 2 with 20 degrees, respectively. Link 1 had rotatory motion perpendicular to the ground, so the tip position was unaffected. The latter half was supplied with 0 degrees which restored the manipulator to the original position. The manipulator inputs were as shown in Figure 2. For the purposes of comparison, the input pattern shown was maintained.

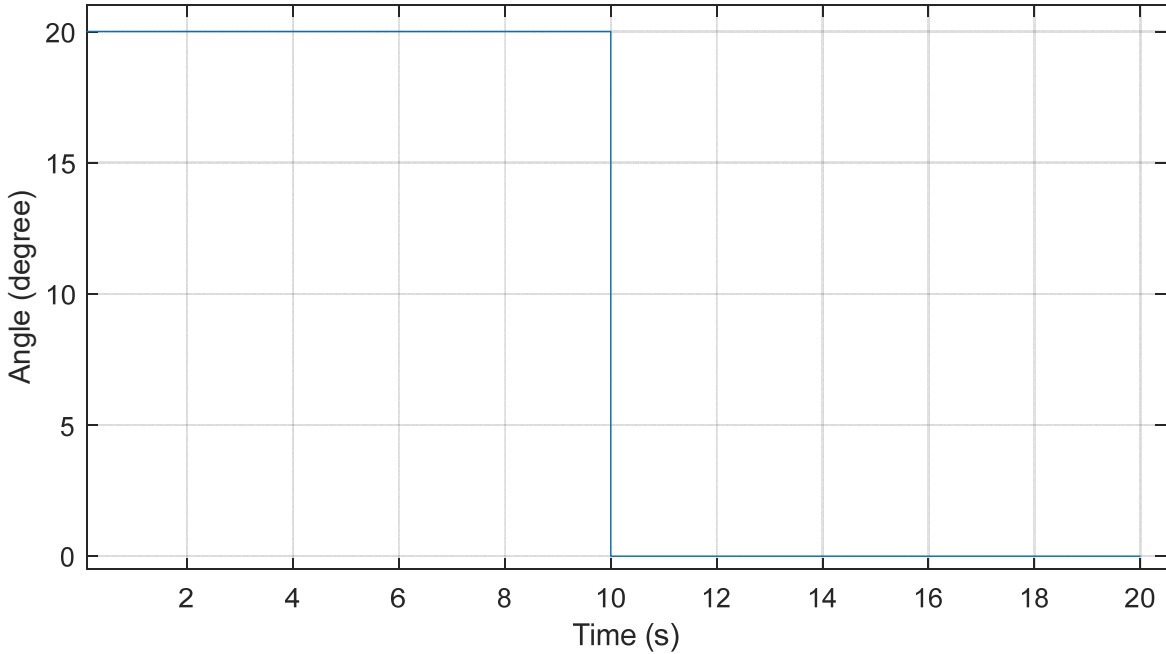

**Figure 2.** Desired input angles of the manipulator.

### *2.2. Linear and Nonlinear Model*

One of the prerequisites for controller design is the knowledge of the plant. Thus, the accurate modelling of the plant is indispensable. Mathematical modelling involving the application of Lagrangian or the Euler–Newtonian formulation has been proposed [22,23]. These methods are tedious and prone to errors with a growing number of links and joints. An alternative is the use of system identification and symbolic modelling.

Symbolic modelling involves the use of computer applications to model the plant. A mathematical representation of the plant is acquired as state space or in differential algebraic equations. In the research, we utilized Maplesim to simulate the actual plant presented in Figure 1. Figure 3 shows the virtual model developed in the Maplesim environment. This involves assembling various blocks and components that best fit the actual manipulator, i.e., flexible beams, joints, rigid bodies, actuators, and motors, amongst others. The physical parameters of these blocks are iteratively changed until there is an agreement between the simulation and the physical model.

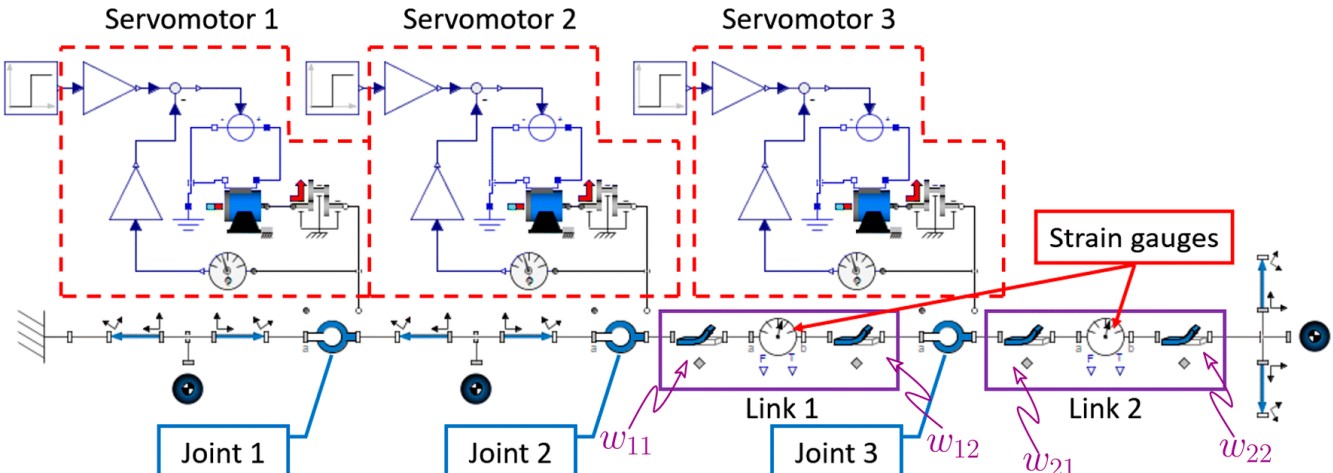

**Figure 3.** Maplesim model of the manipulator.

Maplesim software is good at modeling but not at the analysis and synthesis of the controller. For this reason, after successfully running the model in Maplesim, the Matlab/Simulink block was extracted and exported to Matlab. The extraction of the model involves Maplesim's inbuilt function generator. In the paper, the resulting model had states as in Table 2 below:

**Table 2.** States of Maplesim model.

| State No. | State | Variable | Description |
|:---:|:---:|:---:|:---|
| 1 | $x_1(t)$ | $i_1(t)$ | Armature current of motor 1 |
| 2 | $x_2(t)$ | $w_{11}(t)$ | Link flexure of link 1, part I |
| 3 | $x_3(t)$ | $\dot{w}_{11}(t)$ | Time derivative of link flexure of link 1, part I |
| 4 | $x_4(t)$ | $w_{12}(t)$ | Link flexure of link 1, part II |
| 5 | $x_5(t)$ | $\dot{w}_{12}(t)$ | Time derivative of link flexure of link 1, part II |
| 6 | $x_6(t)$ | $w_{21}(t)$ | Link flexure of link 2, part I |
| 7 | $x_7(t)$ | $\dot{w}_{21}(t)$ | Time derivative of link flexure of link 2, part I |
| 8 | $x_8(t)$ | $w_{22}(t)$ | Link flexure of link 2, part II |
| 9 | $x_9(t)$ | $\dot{w}_{22}(t)$ | Time derivative of link flexure of link 2, part II |
| 10 | $x_{10}(t)$ | $\theta_1(t)$ | Angle, joint 1 |
| 11 | $x_{11}(t)$ | $\dot{\theta}_1(t)$ | Velocity, joint 1 |
| 12 | $x_{12}(t)$ | $\theta_2(t)$ | Angle, joint 2 |
| 13 | $x_{13}(t)$ | $\dot{\theta}_2(t)$ | Velocity, joint 2 |
| 14 | $x_{14}(t)$ | $\theta_3(t)$ | Angle, joint 3 |
| 15 | $x_{15}(t)$ | $\dot{\theta}_3(t)$ | Velocity, joint 3 |
| 16 | $x_{16}(t)$ | $i_3(t)$ | Armature current of motor 3 |
| 17 | $x_{17}(t)$ | $i_2(t)$ | Armature current of motor 2 |

where $i_j$ denotes the armature current to the servomotor driving joint ($j$ = 1,2,3) and $\theta_j$ and $\dot{\theta}_j$ are the angle and velocity of joint ($j$ = 1,2,3), respectively, whereas ($w_{11}$; $w_{12}$) and ($w_{21}$; $w_{22}$) and their time derivatives ($\dot{w}_{11}$; $\dot{w}_{12}$) and ($\dot{w}_{21}$; $\dot{w}_{22}$) denote the flexure variable for links 1 and 2, respectively.

In the system identification, the objective was to formulate the state space model of the manipulator via experimentation. We used MATLAB's System Identification Toolbox to perform system identification using the canonical variate analysis approach with the input and output data generated by the experiment. The input was the displacement of every joint as recorded by the encoder module while the output was the tip trajectory as recorded by the OptiTrack camera system. The state space model was employed since the system contained many inputs and outputs. In this case, the number of states was chosen as ten. The equation of the state of the identification system is represented by Equation (1) as the direct transfer term D was set to 0.

$$\dot{x} = Ax + Bu$$
$$y = \mathbf{C}x$$
(1)

$x$, $y$, and $u$ are given as:

$$x = \{x_1\ x_2\ x_3\ x_4\ x_5\ x_6\ x_7\ x_8\ x_9 x_{10}\}^T$$
$$y = \{y_1\ y_2\ y_3\}^T$$
$$u = \{u_1\ u_2\ u_3\}^T$$

The $x$, $y$, and $z$ coordinates of the tip trajectory are shown by $y_1, y_2, y_3$ which indicate the $x$, $y$, $z$, and the displacements of the input angles to joint 1, joint 2, and joint 3 are indicated by $u_1, u_2, u_3$, respectively. To obtain Equation (2), we first differentiated the lower expression of Equation (1). Equation (3) was thereby obtained by substituting (1) into (2).

$$\dot{y} = \mathbf{C}\dot{x} \tag{2}$$

$$\dot{y} = \mathbf{C}\dot{x} = \mathbf{C}(\mathbf{A}x + \mathbf{B}u) = \mathbf{C}\mathbf{A}x + \mathbf{C}\mathbf{B}u \tag{3}$$

We introduced the relative degree $n$ of the output matrix $\mathbf{C}$, where $\mathbf{C}\mathbf{A}^l\mathbf{B} = 0$, $\forall l < r - 1$, and differentiated Equation (3) repeatedly for $r$ times. Assuming that $\mathbf{C}\mathbf{A}^r\mathbf{B} \neq 0$, we obtained

$$y^{(r)} = \mathbf{C}\mathbf{A}^r x + \mathbf{C}\mathbf{A}^{r-1}\mathbf{B}u \tag{4}$$

The further transformation of (4) was performed to obtain (5) which is the input vector $u$.

$$u = \left(\mathbf{C}\mathbf{A}^{r-1}\mathbf{B}\right)^{-1}\left[y^{(r)} - \mathbf{C}\mathbf{A}^r x\right] \tag{5}$$

Substituting (5) into (1), we obtained

$$\begin{aligned}
\dot{x} &= \mathbf{A}x + \mathbf{B}\left(\mathbf{C}\mathbf{A}^{r-1}\mathbf{B}\right)^{-1}y^{(r)} - \mathbf{B}\left(\mathbf{C}\mathbf{A}^{r-1}\mathbf{B}\right)^{-1}\mathbf{C}\mathbf{A}^r x \\
&= \left\{\mathbf{A} - \mathbf{B}\left(\mathbf{C}\mathbf{A}^{r-1}\mathbf{B}\right)^{-1}\mathbf{C}\mathbf{A}^r\right\}x + \mathbf{B}\left(\mathbf{C}\mathbf{A}^{r-1}\mathbf{B}\right)^{-1}y^{(r)}
\end{aligned} \tag{6}$$

In addition, from (5)

$$u = -\left(\mathbf{C}\mathbf{A}^{r-1}\mathbf{B}\right)^{-1}\mathbf{C}\mathbf{A}^r x + \left(\mathbf{C}\mathbf{A}^{r-1}\mathbf{B}\right)^{-1}y^{(r)} \tag{7}$$

From these, the inverse state space system can be expressed as follows.

$$\begin{cases} \dot{\hat{x}} = \hat{\mathbf{A}}\hat{x} + \hat{\mathbf{B}}y^{(r)} \\ u = \hat{\mathbf{C}}\,\hat{x} + \hat{\mathbf{D}}y^{(r)} \end{cases} \tag{8}$$

where:

$$\hat{\mathbf{A}} = \mathbf{A} - \mathbf{B}(r\mathbf{B})^{-1}\mathbf{C}\mathbf{A}^r \tag{9}$$

$$\hat{\mathbf{B}} = \mathbf{B}\left(\mathbf{C}\mathbf{A}^{r-1}\mathbf{B}\right)^{-1} \tag{10}$$

$$\hat{\mathbf{C}} = -\left(\mathbf{C}\mathbf{A}^{r-1}\mathbf{B}\right)^{-1}\mathbf{C}\mathbf{A}^r \tag{11}$$

$$\hat{\mathbf{D}} = \left(\mathbf{C}\mathbf{A}^{r-1}\mathbf{B}\right)^{-1} \tag{12}$$

and $\hat{x}$ is the state variable, $u$ is the output variable, and $y^{(r)}$ is the input variable of the inverse state space model depicted in Figure 4 below.

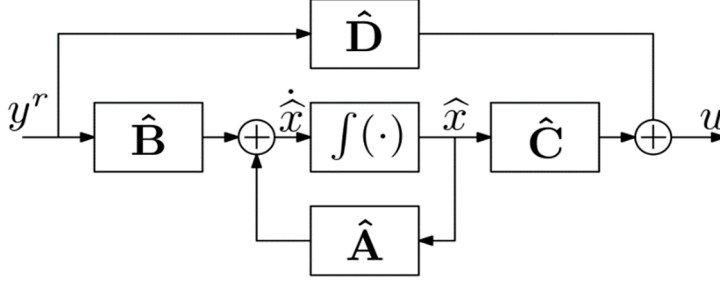

**Figure 4.** State space representation of the inverse system.

The theory of passivity control of a multilink flexible manipulator was developed by Shimizu and M. Sasaki [24]. The strain detected at the flexible link's root is multiplied by a constant gain *k* in this control method, and the resulting signal is utilized to change the control law as shown in Figure 7. The authors in [25] showed that the approach may dampen link vibrations to a satisfactory degree. They further proved that the resultant closed loop is asymptotically stable using an analytical approach.

The identified model and the nonlinear model derived through Maplesim were utilized to validate the agreement of the modeling with the experiment data. The validation results are presented in Figures 4 and 5 below. Figure 5a–c shows the validation of the joint angles while Figure 6a–c shows the validation of the strain information in the nonlinear, linearized model as well as the actual experiment results. From these, we note the agreement of the data and can, thus, deduce that the linear model captures the dynamics of the actual manipulator accurately.

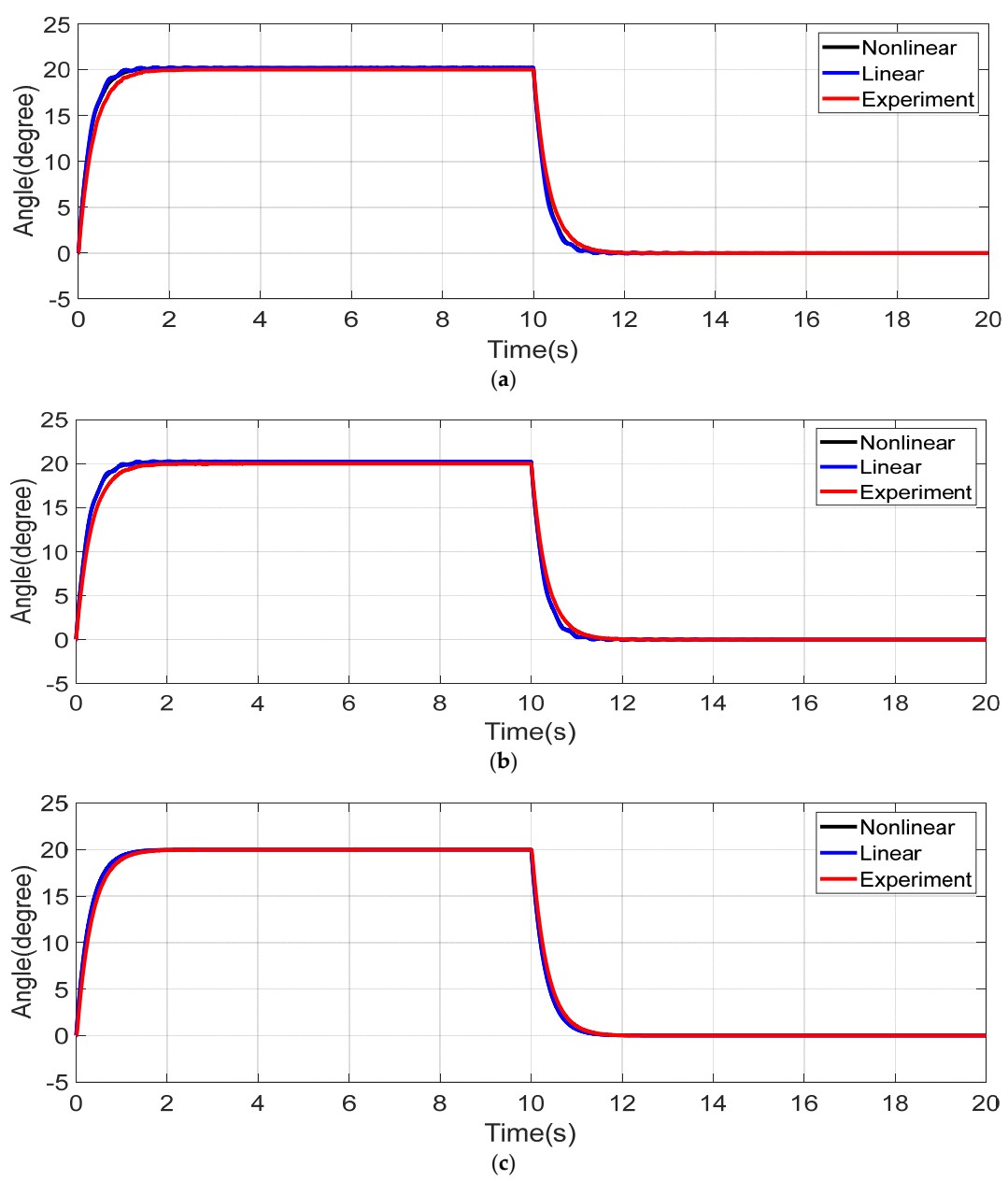

**Figure 5.** Validation of linearized model from joint angles. (**a**) Joint 1, (**b**) Joint 2, (**c**) Joint 3.

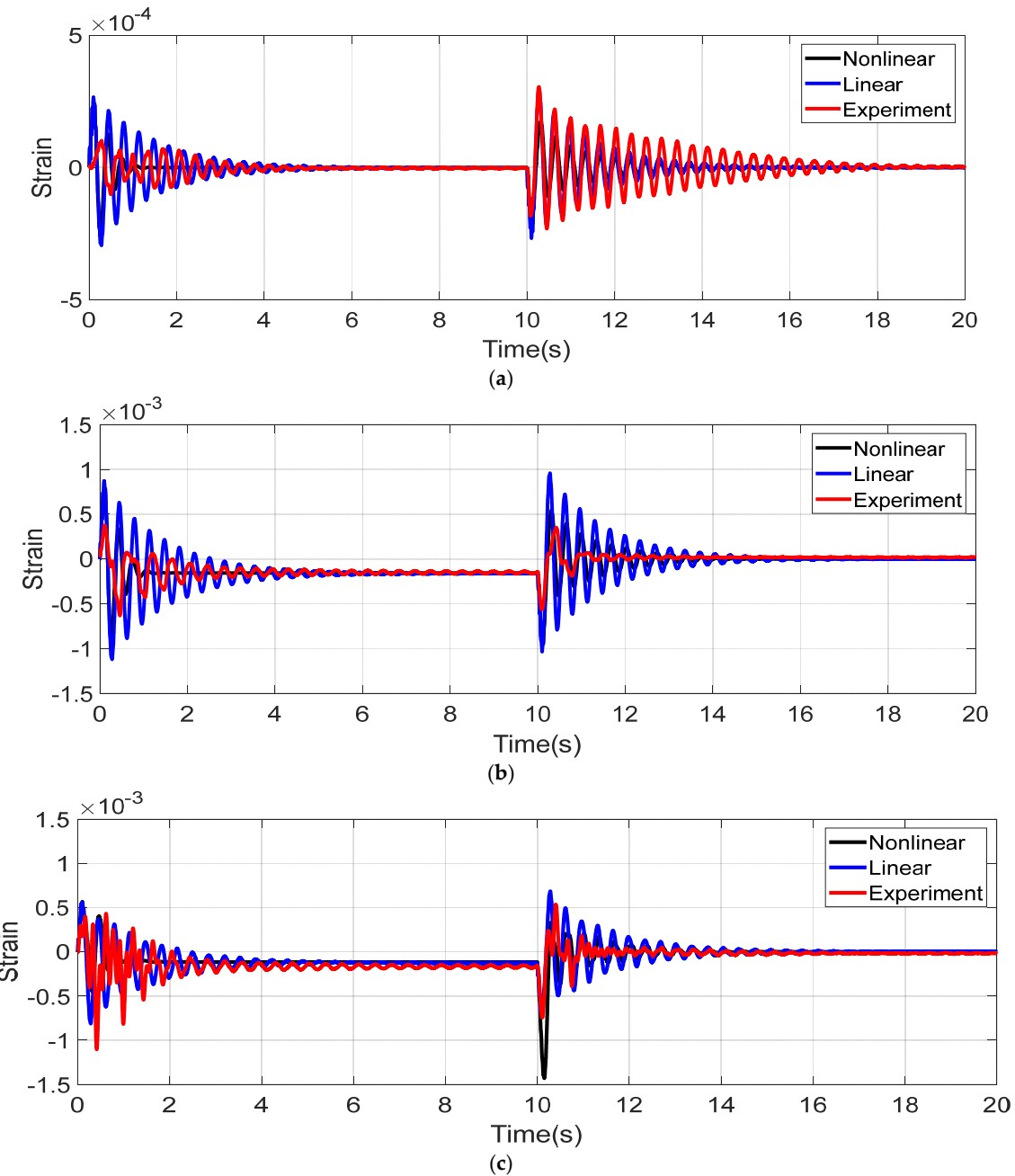

**Figure 6.** Validation of linearized model from strain data. (**a**) Torsion, (**b**) Link 1 in plane, (**c**) Link 2 in plane.

We utilized the linear model to develop the inverse model in Matlab. Inverse control systems are very popular in diverse fields, especially for system analysis; [1,2] are examples of articles on inversion-based system analysis in railway monitoring. From a literature review, inversion can be broadly categorized as right and left inverse [15]. The left inverse is where the plant is connected in series with the inverse at the output stage, while the right inversion is connected such that the inverse model provides the necessary input to the plant. The left inversion seeks to reproduce the input to the plant and is popular in fault detection, while right inverse reproduces the necessary input to the plant for the desired plant output. To this end, this paper utilized the right inversion model as a feedforward control scheme. The derivation and further details of the inversion can be found in our previous research in [26,27].

The inverse system was configured as shown in Figure 7. In the figure, the intended joint angles are sent into the inverse feedforward controller, whose output is fed to the plant to recreate the correct angle. This might result in extremely high and unsafe operational speeds. To avoid this, low-pass filters are used to feed appropriate angles into the inverse feedforward controller. Vibrations are reduced even more by monitoring link vibrations and transmitting them back as shown.

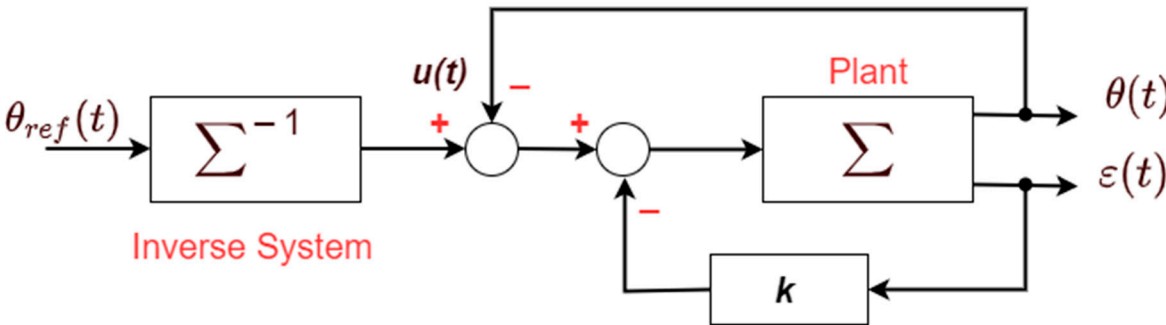

**Figure 7.** Two-degrees-of-freedom inversion system generated by system identification.

Furthermore, as the objective of this study was a reduction in vibration, we designed filters to adapt the controller in real time. Adaptive notch filters are used in applications that automatically detect and remove unknown sine waves superimposed on broadband signals. In this paper, this wide-band signal was regarded as steady distortion, and an unknown sine wave was regarded as a vibration component; the signal from which the vibration component was removed was subtracted from the original signal to realize the steady-state distortion removal. For stability and speed, in this paper, we opted to utilize the simplified lattice algorithm (SLA) as described below.

In the SLA, better convergence than the gradient approach may be accomplished by implementing an adaptive notch filter with a normalized grid type and employing the structure's state variables in an adaptive algorithm. Furthermore, the filter stability is always ensured since the peak gain is normalized.

The setup for the notch filter controller model showing the design of a second-order IIR adaptive notch filter based on the SLA proposed by P.A. Regalia, discussed in [28], is shown in Figure 8. The output of the filter is described by the relation below.

$$H(z) = \frac{1 + sin\theta_2}{2} \times \frac{1 + 2\sin\theta_1(n)z^{-1} + z^{-2}}{1 + \sin\theta_1(n)(1 + \sin\theta_2)z^{-1} + \sin\theta_2 z^{-2}} \tag{13}$$

From Equation (13) above, $\theta_1$ and $\theta_2$ determine the notch frequency and notch bandwidth, respectively. In this case, $f_N$ and $f_s$ are the notch and sampling frequency, respectively, and $B$ is the notch width. From the above, we can express $\theta$ as Equations (10) and (11) below. Note that $0 < B < \pi/2$.

$$sin\theta_2 = \frac{1 - \tan(B/2)}{1 + \tan(B/2)} \tag{14}$$

$$\theta_1(n) = 2\pi f_N/f_s - \frac{\pi}{2} \tag{15}$$

Equation (14) represents the bandwidth of the system, and as such, smaller values of $B$ will have a corresponding narrow notch width. In this work, a constant $B$ of 0.1 was adopted.

From Equation (11) above, the SLA updates coefficients to determine the unknown frequency (notch frequency, $f_N$) using Equation (12) as follows.

$$\theta_1(n + 1) = \theta_1(n) - \mu(n)u_1(n)e(n) \tag{16}$$

where $\mu(n)$ is the adaptation step size, $u_1(n)$ is the state variable of the filter, and $e(n)$ is the output error. The filtered regressor signal is given by Equation (13) as follows.

$$x(n) = \frac{z^{-1}\cos\theta_2\cos\theta_1}{1 + \sin\theta_1(n)(1 + \sin\theta_2)z^{-1} + \sin\theta_2 z^{-2}} \tag{17}$$

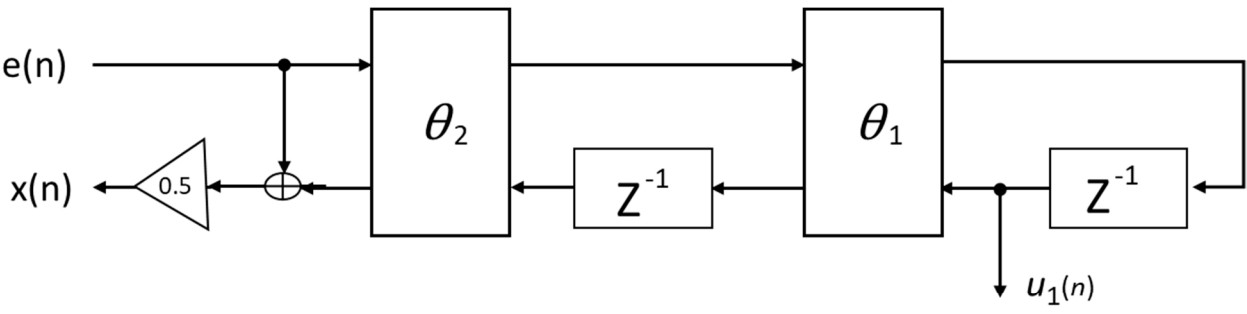

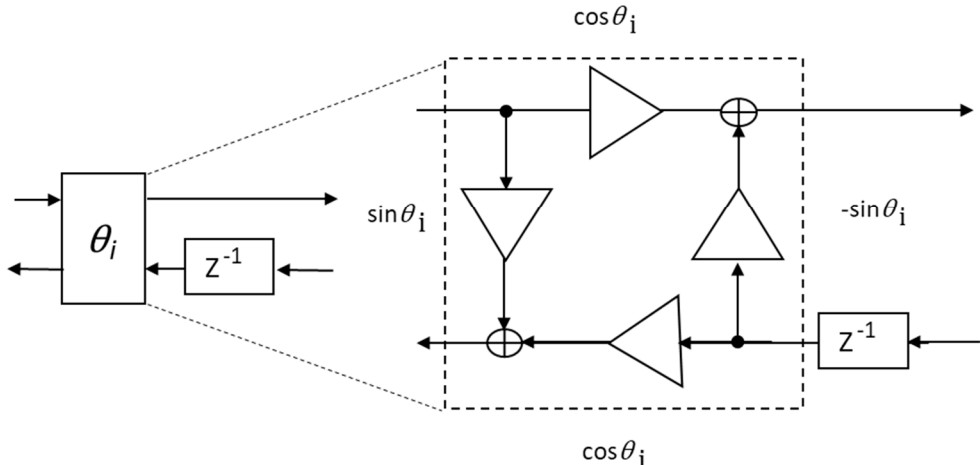

**Figure 8.** Block diagram of the adaptive notch filter in use.

From Equation (12), if the bandwidth is held constant, and the gradient $\partial E[e^2(n)]/\partial\theta_1$ is not a recursive minimization of the output error but it is a globally stable unbiased parameter identifier in the case of single or multiple sinusoids when the bandwidth is narrow. $\mu(n)$ is defined in Equation (14) where $\lambda$ is the "forgetting factor" and $k$ is the step size in consideration. The values of $\mu_0$ and $\lambda$ are determined by trial and error, and for experiments in this paper, both were taken to be equal to 1.0.

$$\mu(n) = \frac{\mu_0}{\sum_{k=0}^{n}\lambda^{n-k}[u_1(n)]^2} \tag{18}$$

The vibration's resonance frequency, which is taken to be the notch frequency $f_N$ above, is determined from strain data. Thus, the input to the notch filter $e(n)$ is the strain feedback data resulting from manipulator movement. We designed the system shown in Figure 9 in Simulink to perform adaptive filtering with the Maplesim-generated model as the plant.

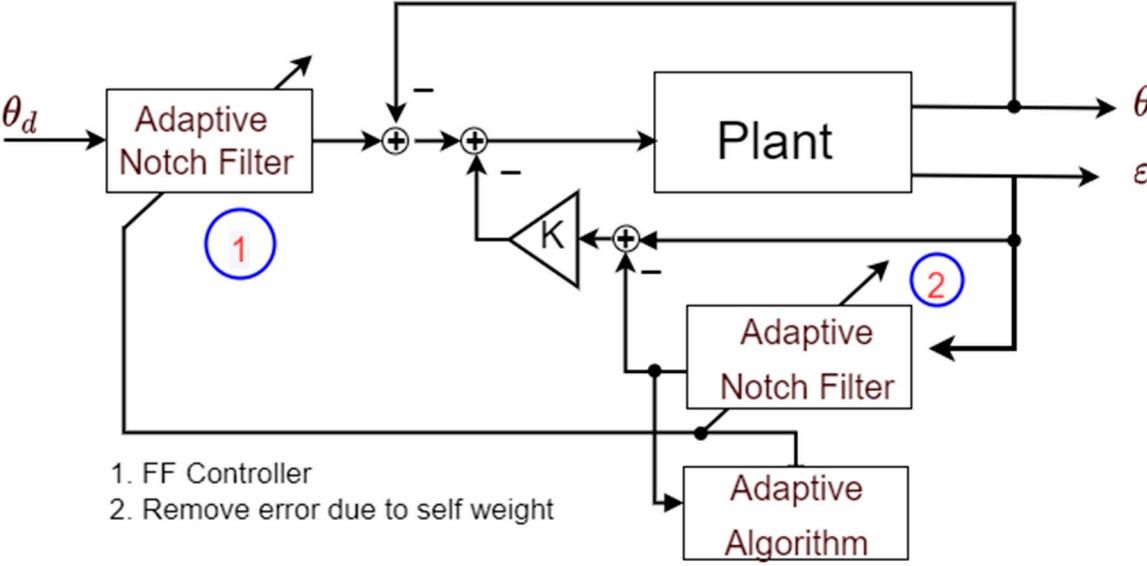

**Figure 9.** Block diagram of the adaptive notch filter feedforward control system.

The input strain is passed through the two stages of the adaptive notch filter for the FF controller and steady-state distortion correction, labeled as 1 and 2 in Figure 9. The developed notch-filtering model was exported to the dSPACE workspace to compare with the inverse model in Section 2.1 through experimentation.

*2.3. Verification of Control Using the Designed Controller*

The two designed systems were exported to the actual manipulator for the verification of performance. The model obtained from the system identification and inversion in Section 2.1, herein referred to as the inverse model, and the Maplesim-simulated model, the notch filter, were exported from Simulink to the dSPACE DSP board as the controllers. The joint angle control law similar to Figure 2 was adopted. Data were recorded for both cases in dSPACE and exported to Matlab to compare the controllers. The results are described in Section 3.

**3. Experimental Results and Discussion**

The experiments presented in this paper involved moving the three joints to +20°, such that the manipulator swept about a 3D space for 10 s and returned the manipulator to its initial vertical posture for 10 more seconds. Figure 10 shows the joint angles for joints 1, 2, and 3.

From the figure, it can be seen that the joints' angles traced similar trajectories for both the filtered inverse controller and the notch filter feedforward controller in terms of the rise time and steady-state response.

Figure 10 shows the link strain for torsion for link 1 and link 2. From the strain data, vibration suppression can be confirmed in both methods. Particularly, the notch filter control was able to suppress the vibration of the first movement within 2 s which was similar to the inverse system. In the second movement at 10 s, the performance of the notch filter was confirmed with the inverse system reporting residual vibration up to 20 s.

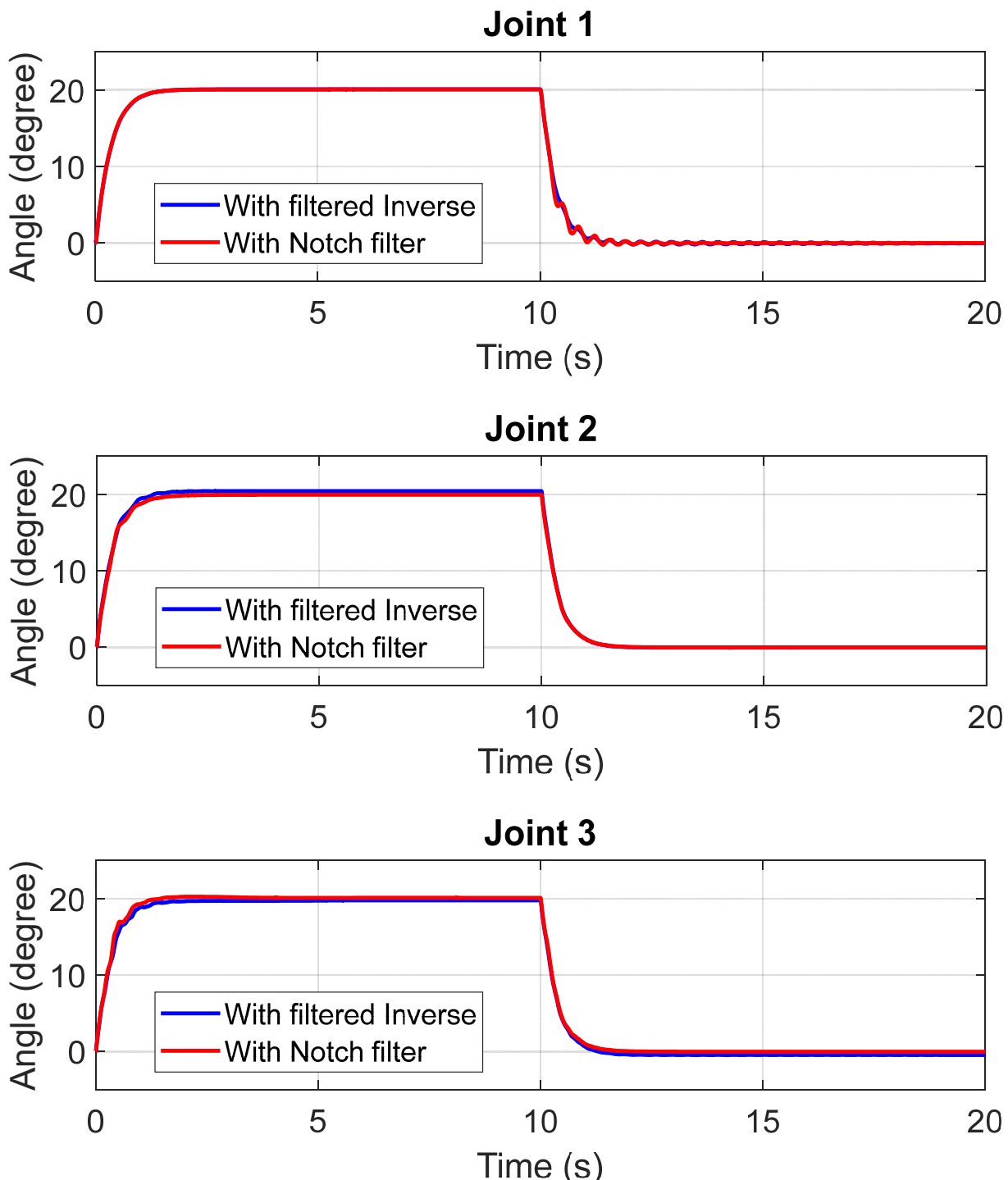

**Figure 10.** Joint angles for filtered and inverse model.

From the figure, it can be seen that the performance of the notch filter feedforward controller was superior to that of the filtered inverse controller in terms of amplitudes and the settling time. This was due to the adaptive nature of the notch filter employed in this work relative to the fixed nature of the filtered inverse controller. For torsion, severe strain can be seen in the period from 10 to 20 s. This was due to the coupling of torsional strain from link 2 to link 1 while the manipulator was in its vertical posture.

Figure 11 shows the comparison of link vibration spectral densities between the filtered inverse controller and the adaptive notch filter controller. The blue line shows the results of the two-degrees-of-freedom control system with a filtered inverse controller, whereas

the red line shows the results of the control system with an adaptive notch filter. The notch filter control cut the resonant frequency components by over 80% as compared to the inverse controller.

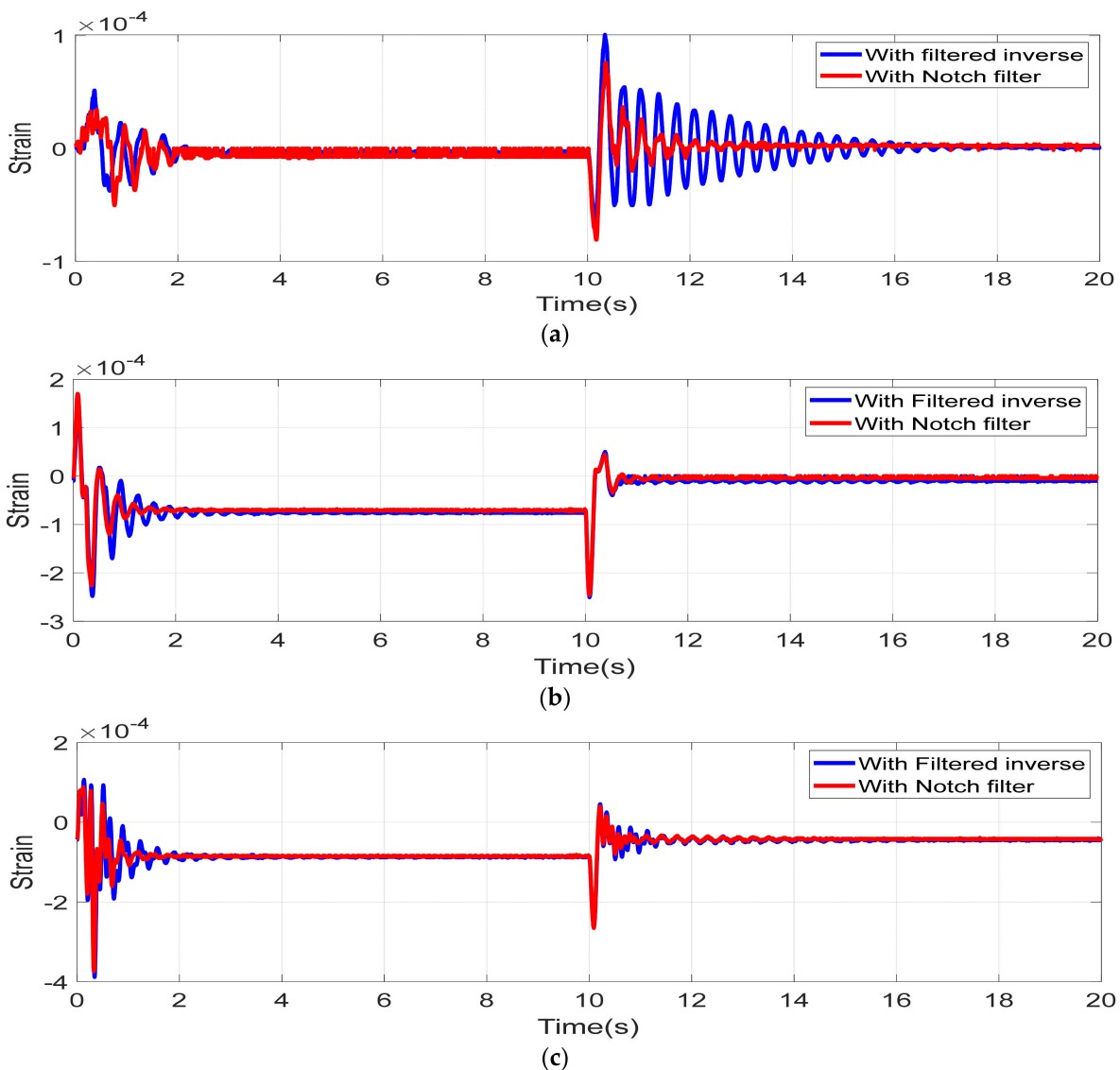

**Figure 11.** Link strain. (**a**) Torsion, (**b**) Link 1 in plane, (**c**) Link 2 in plane.

From Figure 12, it can be seen that the adaptive notch filter controller had superior performance to the filtered inverse controller. The superiority of the adaptive filter stems from its ability to adapt to the vibration frequencies and act accordingly to attenuate these frequencies from being fed back to the manipulator. On the other hand, the effectiveness of the fixed filtered inverse controller in removing residual vibration frequencies from the control law was poor. There still remained 3 Hz residue vibrations in the inverse controlled system. The individual controllers dampened different frequency ranges, with the hybrid performing better than the individual controllers.

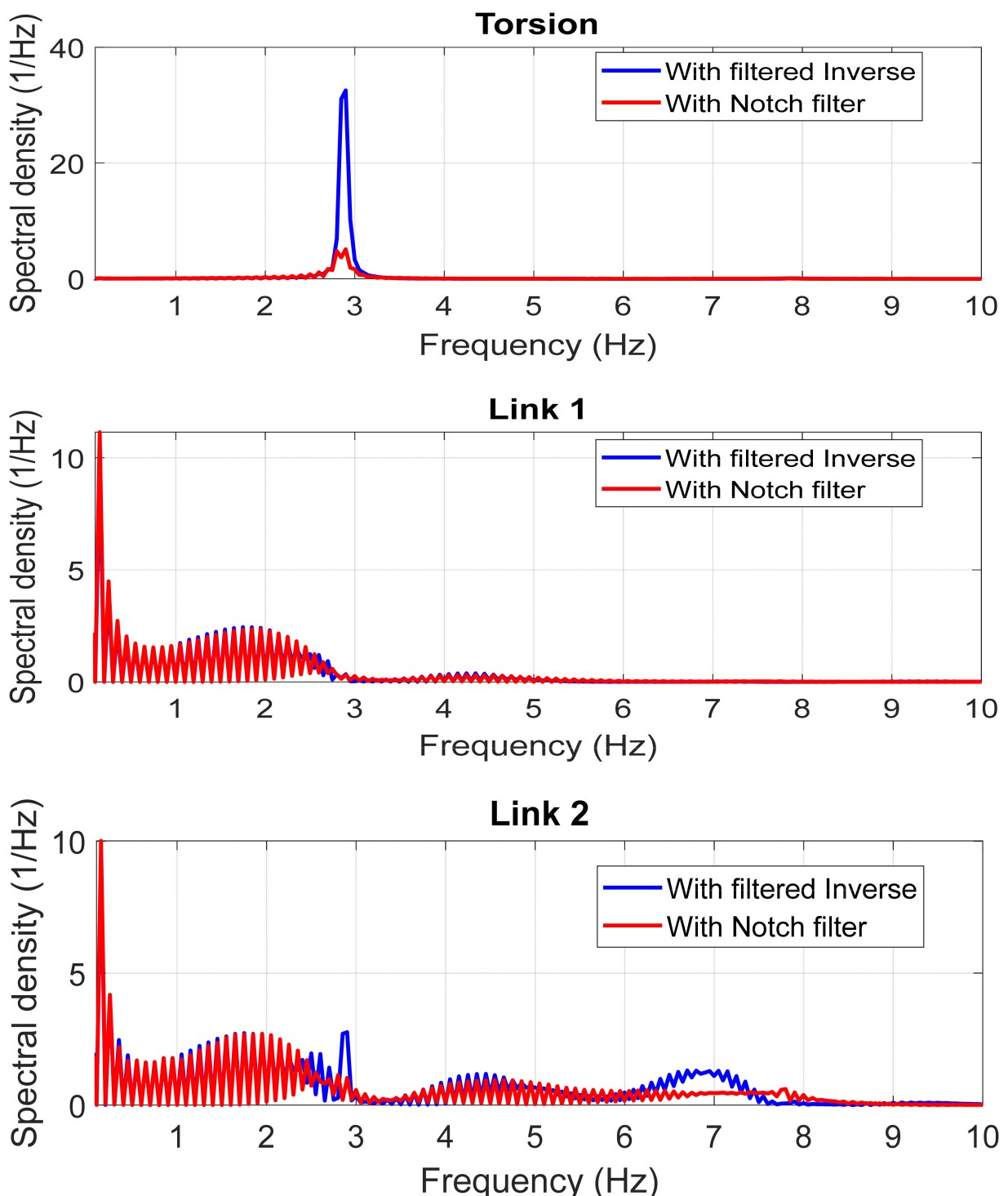

**Figure 12.** Strain spectral power density.

## 4. Conclusions

This paper presented a comparative study of two types of two-degrees-of-freedom controller comprising a filtered inverse controller and an adaptive notch controller in the feedforward path, both augmented with a direct strain feedback controller. From the results of the link strain in the torsional and transverse senses, the performance of the two-degrees-of-freedom controller comprising of the adaptive notch controller and direct strain feedback controller was superior to the controller involving the filtered inverse control. This is attributed to the fact that with the adaptive notch filter, it was possible to adaptively

filter out link vibration in the control law, a feature that was missing in the filtered inverse controller with two degrees of freedom. From the results of the FFT spectrum of strain, the proposed control had an improvement of over 80% in reducing the strain power of the resonant frequency.

In addition, it can be seen that although the two-degrees-of-freedom controller involved feedforward controllers, the joint angles settled to the desired joint angles for both controllers. Further research will be conducted to verify the performance of the system in varying loads and speeds.

**Author Contributions:** Conceptualization, M.S. and W.N.; methodology, W.N.; software, J.M. and A.S.A.D.; validation, M.S., J.M., W.N. and A.S.A.D.; formal analysis, W.N.; investigation, J.M.; resources, M.S.; data curation, A.S.A.D.; writing—original draft preparation, M.S.; writing—review and editing, J.M.; visualization, W.N. and A.S.A.D.; All authors have read and agreed to the published version of the manuscript.

**Funding:** This research received no external funding.

**Data Availability Statement:** No new data were created or analyzed in this study. Data sharing is not applicable to this article.

**Acknowledgments:** This work is partially supported by Grants-in-aid for Promotion of Regional Industry-University-Government Collaboration from Cabinet Office, Japan.

**Conflicts of Interest:** The authors declare no conflict of interest.

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
