# Peer review of "Adaptive Notch Filter in a Two-Link Flexible Manipulator for the Compensation of Vibration and Gravity-Induced Distortion"

_vibration, doi:10.3390/vibration6010018_

Round 1

Reviewer 1 Report

This paper presents a flexible manipulator control approach using an inverse feedforward controller and an adaptive notch filter with a direct strain feedback controller. The results confirmed the feasibility of the approach, characterized by minimal transient vibrations and a quick settling of the end-effector. The manuscript is well organized but requires minor revision for publication. The following comments are suggested.

1. The formulas in lines 202-206 of the manuscript should be numbered.

2. The black lines in Figures 3-4 are difficult to distinguish. Please use different line types to distinguish them.

3. The order of magnitude difference between the ordinates of Figure 4 and Figure 9 reaches 10^4. Please explain why.

4. Inverse analysis of systems has been applied in many fields, such as railway monitoring [1-2], authors should mention such applications in the introduction Section.

[1] X. Xiao, X. Xu, W. Shen. Simultaneous identification of the frequencies and track irregularities of high-speed railway bridges from vehicle vibration data. Mech. Syst. Signal Process, 152 (2021) 107412.

[2] X. Xiao, W. Shen, X. He, Track irregularity monitoring on high-speed railway viaducts: A novel algorithm with unknown input condensation, Journal of Engineering Mechanics-ASCE, 147(6) (2021) 04021029.

Reviewer 2 Report

The paper treats the control of a 2-link manipulator through a novel control law. The paper is clear in its aims. I suggest to improve the description of the model because, e.g., there are variables not clearly defined (e.g., \hat{x}), the way to derive some formulas are not clear (e.g., how to derive (4) where I would expect to find \dot{u}?).

What is the ratio between the sampling frequency used and the maximum frequency value considered in the system model?

Please, also correct typos and improve figures with multiple time functions (e.g., figure 3) because sometimes it is hard to see all the lines. Finally, use SI units and symbols (e.g., seconds must be expressed as s in figures and tables).

Round 2

Reviewer 2 Report

The answer to my point 2 is truncated in the pdf I have. Thus, can you please answer again? Because I cannot see your previous answer.

As for my point 3, I see that in the new plots still SI units are not used. As an example, seconds must be indicated as s. Or again, spectral density has no units (is it 1/Hz?).

Author Response

Response to reviewer’s comments - Round 2
Reviewer #2
1. What is the ratio between the sampling frequency used and the maximum frequency value considered in
the system model?
Response:
The sampling frequency of the experimental and simulation setup was 1kHz whereas the maximum frequency value considered in the system model is 10Hz as seen from the strain spectral power density.
Nyquist rate requires that the sampling frequency be at least twice the highest frequency under consideration which is 20Hz. In this case, 1kHz is high enough to bring forth all the information required to
validate the models.
2. Please, also correct typos and improve figures with multiple time functions (e.g., figure 3) because sometimes it is hard to see all the lines. Finally, use SI units and symbols (e.g., seconds must be expressed as
s in figures and tables).
Response:
The authors gladly and carefully revised the draft article, corrected all typos and harmonized all units.
The SI units were harmonized.

Reviewer 3 Report

no comment

Author Response

(The authors gave the same response as above.)
